

# An empirical model of nitric oxide in the upper mesosphere and lower thermosphere based on 12 years of Odin-SMR measurements

Joonas Kiviranta[1], Kristell Pérot[1], Patrick Eriksson[1], and Donal Murtagh[1]

[1]Chalmers University of Technology, Department of Earth and Space Sciences, SE-412 96, Gothenburg, Sweden

*Correspondence to:* Joonas Kiviranta (joonas.kiviranta@chalmers.se)

**Abstract.** Nitric Oxide (NO) is produced by solar photolysis and auroral activity in the upper mesosphere and lower thermosphere region and can, via transport processes, eventually impact the ozone layer in the stratosphere. This work uses measurements of NO taken between 2004 and 2016 by the Odin Sub Millimetre Radiometer (SMR) to build an empirical model which links the prevailing solar and auroral conditions with the measured number density of NO. The measurement data are averaged daily and sorted into altitude and magnetic latitude bins. For each bin, a multivariate linear fit with five inputs, the planetary K-index, solar declination, and the F10.7cm flux, and two newly devised indices which take the planetary K-index and the solar declination as inputs in order to take NO created on previous days into account, constitutes the link between environmental conditions and measured NO. This results in a new empirical model, SANOMA, which only requires the three indices to estimate NO between 85 km-115 km and 80° S-80° N in magnetic latitude. Furthermore, this work compares the NO calculated with SANOMA and an older model, NOEM, with measurements of the original SMR-dataset, as well as measurements from four other instruments: ACE, MIPAS, SCIAMACHY, and SOFIE. The results suggest that SANOMA can capture roughly 31-70% of the variance of the measured datasets near the magnetic poles, and between 16-73% near the magnetic equator. The corresponding values for NOEM are 12-38% and 7-40%, indicating that SANOMA captures more of the variance of the measured datasets than NOEM. The simulated NO for these regions was on average 20% larger for SANOMA, and 78% larger for NOEM, than the measured NO. Two main reasons for SANOMA outperforming NOEM are identified. Firstly, the input data (Odin SMR NO) for SANOMA spans over 12 years, while the input data for NOEM from the Student Nitric Oxide Experiment (SNOE) only covers 1998-2000. Additionally, some of the improvement can be accredited to the introduction of the two new indices, since they include information of auroral activity on prior days which can significantly enhance the number density of NO in the MLT during winter in the absence of sunlight. As a next step, SANOMA could be used as input in chemical models, as apriori information for the retrieval of NO from measurements, or as a tool to compare Odin SMR NO with other instruments. SANOMA and accompanying scripts are available on odin.rss.chalmers.se

## 1 Introduction

Nitric oxide (NO) is a reactive free radical, and together with Nitrogen dioxide ($NO_2$), constitutes the $NO_x$ compounds. Whereas tropospheric $NO_x$ may originate from both natural sources, such as forest fires, and anthropogenic sources, such as combustion engines (Wallace and Hobbs, 2006), NO in the mesosphere and lower thermosphere (MLT, ~50-150 km) has a



purely natural origin. Knowledge about NO in this region is of great importance, because it can affect the atmospheric layers below. To understand how, we need basic knowledge on the chemical reactions which create and destroy NO. In the MLT, NO can be produced by two mechanisms: either direct radiation from the sun in the form of solar-soft x-rays ($8 \leq \lambda \leq 12\,\text{Å}$) or energetic particle precipitation (EPP) (Barth et al., 1999; Sinnhuber et al., 2012). These particles can include electrons, protons,

or heavier ions. These may originate directly from the sun, from aurorae and the radiation belts during geomagnetic storms, or from outside of the solar system.

     Previous studies have established that EPP from auroral activity dominates the variation of NO near the magnetic poles, while solar-soft x-rays contribute more near the magnetic equator (Gérard and Barth, 1977; Barth et al., 1999; Sinnhuber et al., 2012). NO in the MLT is created through the reaction between molecular oxygen and an excited nitrogen according to the

equation

$$N(^2D) + O_2 \rightarrow NO + O, \tag{R1}$$

in which N($^2$D) denotes an excited nitrogen atom. Excited nitrogen is created by the two reactions

$$N_2 + e^* \rightarrow N(^2D) + N, \tag{R2}$$

and

$$NO^+ + e \rightarrow N(^2D) + O, \tag{R3}$$

in which e$^*$ indicates an energetic electron, originating from either EPP or solar soft x-rays (Marsh et al., 2004). This implies, that auroral or solar activity is necessary to form NO in the presence of an oxygen molecule. On the other hand, solar light destroys NO in the cannibalistic reaction

$$NO + h\nu \rightarrow N + O, \tag{R4}$$

$$N + NO \rightarrow N_2 + O. \tag{R5}$$

     Therefore the amount of NO is affected by seasonal variation of sunlight. Under sunlight conditions in the MLT region, NO has a chemical lifetime of less than one day, whereas during the polar night in winter, it may persist for several weeks (Minschwaner and Siskind, 1993). This increased lifetime, together with a stable polar vortex, can eventually result in the

descent of NO into the stratosphere where it can partake in catalytic cycles to destroy ozone (Siskind et al., 2000; Pérot et al., 2014). Additionally, the amount of NO influences the thermal balance of the MLT, especially at ultra-violet wavelengths (Richards et al., 1982). These effects highlight the importance of understanding the mechanisms by which solar and auroral activity create and destroy NO.





This study focuses on the effect of solar and auroral activity on the amount of NO in the MLT. Over the past several decades, at least six satellites have measured NO in the MLT region. These include the past instruments, SNOE (Student Nitric Oxide Experiment), SCIAMACHY (Scanning Imaging Absorption spectroMeter for Atmospheric CHartographY), MIPAS (Michelson Interferometer for Passive Atmospheric Sounding), as well as the currently active Odin SMR (Sub Millimetre

Radiometer), SOFIE (Solar Occultation for Ice Experiment), and ACE (Atmospheric Chemistry Experiment) instruments. The limitation of satellite measurements is that they only cover certain locations and periods of time. Yet, many applications, such as chemical models of the upper atmosphere, require information on the amount of NO at any given time or location. To help bridge this gap, a model which connects known environmental conditions, such as auroral activity, with measured NO can help to provide an estimate of NO at any time and place. Such a model can also help validate and constrain poorly resolved or

underdetermined parameters of first principle models.

Marsh et al. (2004) derived an empirical model, NOEM (Nitric Oxide Empirical Model), which calculates the zonally averaged number density of NO on a grid of magnetic latitude and altitude using the Kp-index, solar declination, and the 10.7 cm solar flux as inputs. Section 2.3 describes the Kp-index and 10.7 cm solar flux while Section 4 specifies the parameters of NOEM in more detail. Empirical Orthogonal Function (EOF) analysis of NO measured with SNOE, which operated between

1998-2000 and measured UV-fluorescence scattering of incident solar radiation (Barth et al., 2003), forms the basis for the derivation of NOEM. Bender et al. (2015) and Bermejo-Pantaleon et al. (2011) use the NO calculated with NOEM as a priori in the NO retrievals of the SCIAMACHY and MIPAS instruments respectively. Moreover, NOEM constitutes the upper boundary for the NCAR (National Center for Atmospheric Research) WACCM (Whole Atmosphere Community Climate Model) at 140 km and helps to evaluate the response of WACCM to variability around 100 km in altitude.

However, no study has validated NOEM nor proposed a contending model since its release. This study aims to fill these two gaps by building a new empirical model based on NO measurements in the MLT by Odin SMR for the period 2004-2016. We hypothesize that an empirical model derived from Odin SMR should be more accurate than NOEM because the Odin SMR measurements include a larger range of solar conditions over a period of over 12 years. Furthermore, SNOE measured only daytime NO whereas SMR provides NO measurements during both day and night. This might introduce some discrepancy

between NOEM and the resulting empirical model, since night-time NO is expected to be higher than day-time NO due to its dissociation by sunlight.

This study primarily aims to derive a new empirical model based the 12 years of Odin SMR measurements to calculate NO in the MLT. This new model will be named the SMR Acquired Nitric Oxide Model Atmosphere (SANOMA). Additionally, this study aims to evaluate the performance of both SANOMA and NOEM by comparing simulated NO with measurements

from the independent NO-measuring instruments SOFIE, SCIAMACHY, ACE, and MIPAS.

Section 2 describes the SMR NO data set and space weather indices, while section 3 thoroughly describes the method used to derive SANOMA from the Odin SMR observations. Finally, Section 4 assesses the performance of both NOEM and SANOMA by comparing their simulated NO with measured NO from aforementioned satellites, followed by a discussion in Section 5.



## 2 Data Description

This section outlines the Odin SMR data set which forms the basis for SANOMA. Section 2.1 describes the contents of the original data set of Odin SMR measurements while Section 2.2 presents the steps taken from this data set to the NO data used for the analysis. Finally, Section 2.3 offers an overview of the indices used in this study to describe auroral and solar activity.

### 2.1 Odin SMR

The Odin SMR instrument scans the limb of the atmosphere and has been observing NO thermal emission lines in a band centered around 551.7 GHz since October 2003 (Sheese et al., 2013). Frisk et al. (2003) provide a description of the Odin SMR instrument. Odin orbits the Earth in a sun-synchronous orbit with ascending and descending nodes around 06:00 and 18:00 local time, respectively (Murtagh et al., 2002), and provides near-global coverage between approximately 82° S and 82° N, although some measurements can be located at higher latitudes, as the satellite is turned to point towards the poles when conditions allow for this. From 2003 to 2007, the Odin SMR instrument split its measurement time between aeronomy and astronomy modes. Consequently, during this period the NO mode of the SMR instrument operated a 24 hour period approximately once per month whereas subsequent to 2007, it has measured approximately four 24-hour periods each month. Since the SMR instrument measures microwave emission, the data set also contains both day- and night-time measurements.

Using the measured emission spectra, an inversion algorithm derives the Volume Mixing Ratio (VMR) of NO as a function of altitude for the location of the measurement with an altitude resolution of ~4 km. This inversion technique uses the measured brightness temperatures of the NO emission from the various tangent altitudes of a single limb scan as well as a priori NO climatology to ascertain an estimate for the VMR of NO as a function of altitude. Version 3.0 Level 2 NO VMR constitute the Odin SMR data used in this study.

Only measurements in which the measurement response, a measure of the relative contributions of the measurement and the a priori, exceeds 0.75 are considered for our analysis. Although no study has validated the version 3.0 Level 2 data, Bender et al. (2015) found that version 2.1 SMR NO measurements were consistent with NO measurements from the ACE, MIPAS, and SCIAMACHY instruments.

### 2.2 Daily Zonal Averages

SANOMA will express NO in number density to accommodate chemical models. The VMR at each altitude is converted to number density, molecules/cm$^3$, with the ideal gas law in which the pressure and temperature originate from an apriori background atmosphere. For the analysis, an algorithm calculates the daily averages of NO number density as a function of altitude and altitude-corrected magnetic latitude. Since the original data set contains measurements on geographical coordinates, a Matlab routine based on the IGRF-12 (International Geomagnetic Reference Field) internal field model, converts the geographical latitude to altitude-corrected magnetic latitude (Thebault et al., 2015), denoted with $\Lambda$.

For each measurement day, the NO number density is sorted into bins according to altitude and magnetic latitude. Prior to sorting, each individual measurement is interpolated in altitude with grid points at the centers of the altitude bins. The bins





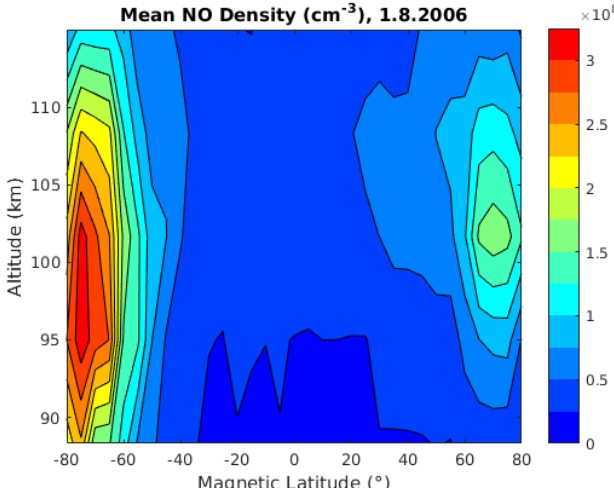

**Figure 1.** Mean NO density in molecules/cm$^3$ for 1.8.2006 calculated from V3.0 Level 2 Odin SMR data.

for altitude run from 85 km to ∼118.33 km with a bin width of ∼6.67 km. This altitude range reflects the area in which the measurement response of the SMR-measured NO typically exceeds 0.75, while the resolution reflects that of the measurement. The bins for magnetic latitude run from -82.5°Λ to 82.5°Λ with a bin width of 5°Λ. Once the algorithm has sorted the individual measurements for the measurement day, it calculates the mean NO number density in each bin. Although the Odin orbit should

ensure near-global coverage, on some days the data covers only a limited range of altitudes and latitudes due to gaps in the measurements. Figure 1 illustrates an example of the average NO number density for the 1st of August 2006. Maximum NO concentrations can be seen around -70°Λ and 70°Λ and reflect the locations of maximum auroral activity. With all measurement days, the resulting NO number density data comprises 442 individual days from January 2004 to April 2016, each containing 33 latitude and 5 altitude bins. Therefore, a 3D-matrix with the dimensions 442×5×33 contains all the necessary information

on NO. Although Odin has continued measuring subsequent to 2016, the used period already contains one solar cycle and is thus deemed sufficient for the analysis in this work.

### 2.3 Space Weather Indices

Since auroral and solar activity create NO in the MLT, proxies which describe these two phenomena constitute key parts of SANOMA. In the search for appropriate proxies, this chapter introduces some of the most common ones used.

Measurements of the irregular variations of the horizontal component of the Earth's magnetic field constitute an auroral activity index, the Kp-index (Menvielle and Berthelier, 1991). It ranges from 0-9 on a quasi-logarithmic scale with higher values corresponding to higher activity. It is derived every 3 hours from a network of measurements stations located between ∼40° N-60° N as well as at ∼40° S geographical latitude. A similar index, the Ap-index is exponentially proportional to the Kp-index. Finally, the Ae-index is another measure for auroral activity and has been suggested to correlate more directly with

NO in the MLT region than the Kp- or Ap-index (Hendrickx et al., 2015). The Ae-index derives from measurement stations





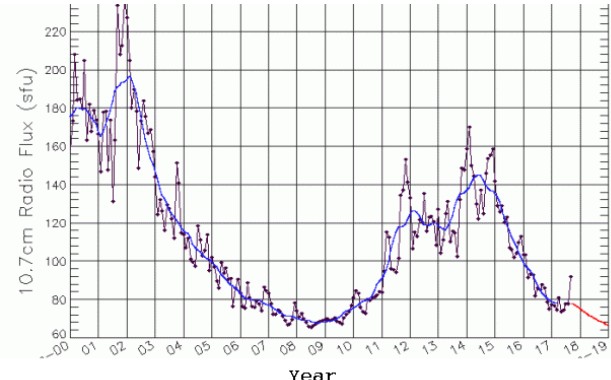

**Figure 2.** F10.7cm flux over time. (NOAA, 2017)

closer to the magnetic poles than the Kp-index (Menvielle et al., 2011). The Kp- and Ap-indices for this study are obtained from the National Oceanic and Atmospheric Administration (NOAA) whereas the Ae-index originates from http://wdc.kugi.kyoto-u.ac.jp/aedir/.

To describe solar activity, the 10.7 cm solar radio flux is among the most widely used indices. It constitutes a proxy for the in-
coming solar soft x-rays and is based on the solar radio emission in a 100 MHz-wide band centered around 2800 MHz (Tapping and Detracey, 1990). This study uses the observed daily mean 10.7 cm from the NOAA. The total radiation flux centered around the Lyman-$\alpha$ line defines an alternative proxy for solar activity and originates from http://lasp.colorado.edu/lisird/data.html. Figure 2 shows the F10.7cm flux over time. Over the measurement period of Odin SMR, solar activity decreased from 2003 to 2009, subsequent to which it rose to reach a maximum in 2014, followed by another decline. In the context of this work, the
solar cycle is integral to understanding radiation-related variations of NO.

## 3  SANOMA

This section describes the method used to derive SANOMA from the original Odin SMR measurements. Section 3.1 presents the evidence on how various environmental conditions are linked to the amount of measured NO and presents the underlying equations of SANOMA. Section 3.2 compares SANOMA as well as an empirical model derived from Odin SMR NO but using
the principle of NOEM, with NO measured with Odin SMR.

### 3.1  Linking Environmental Conditions and Measurements to Form SANOMA

Figure 3 displays the mean NO from the Odin SMR dataset. Distinct maxima of up to $16 \times 10^7$ cm$^{-3}$ around $\pm 70°\Lambda$ can be observed at an altitude of roughly 102 km. Figure 4 shows the time series of the daily means of NO derived from the V3.0 Odin SMR data at the altitude-latitude bin centered around a) 102 km and $-70°\Lambda$ and b) 102 km and $0°\Lambda$. Figure 4a) displays
strong seasonal variability, while the time series in Figure 4b) exhibits long-scale variability which can be associated with the



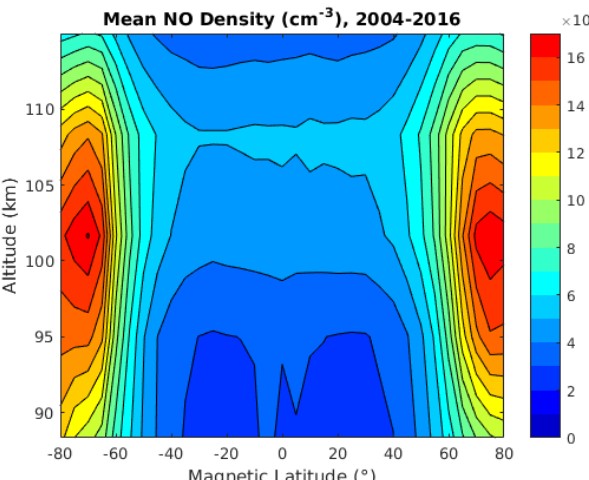

**Figure 3.** Mean NO density of NO measured by Odin SMR from 2004-2016.

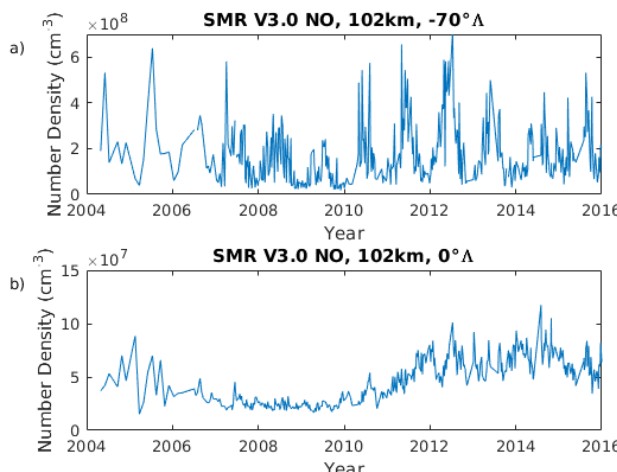

**Figure 4.** a) Daily mean NO density of Odin SMR NO at a) 102 km and -70°$\Lambda$ and b) 102 km and 0°$\Lambda$, from 2004 to 2016.

solar cycle (Baker et al., 2001). The highest peaks in Figure 4a) are most likely produced by high levels of auroral activity. These enhancements are also linked to the seasonal variation, with highest levels during winter, during which NO created in the lower thermosphere is transported downwards in the polar vortex (Pérot et al., 2014).

To investigate the link between auroral activity and the NO number density, Figure 5 zooms in on the year 2015 from the same time series as in Figure 4 along with the daily mean of the a)$K_p$ and b)$10.7cm$ fluxes from one day prior to the measurements. Figure 5a) suggests that the $K_p$ index correlates with the amount of NO at the chosen location. Further, Figure 5b) illustrates the link between tropic NO and the $F10.7cm$ flux.



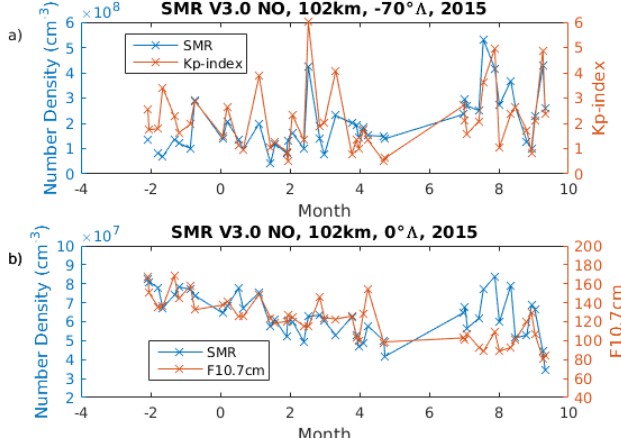

**Figure 5.** a) Daily mean NO derived from Odin SMR at 102km and -70° magnetic latitude with the Kp-index b) 102 km and 0°Λ with the F10.7cm flux.

(Marsh et al., 2004) used the Kp-index and $F10.7cm$ flux as a proxies for auroral and solar activity, respectively. (Marsh et al., 2004) used these two indices as well as solar declination as inputs for a model, NOEM, to calculate the variations of NO in the MLT. Understanding the theory behind NOEM is necessary to have a foundation upon which we build SANOMA. NOEM is based on principle component analysis of two years of measurements from the Student Nitric Oxide Experiment (SNOE) between 1998 and 2000, in which the variation of daily mean NO in time is described using Empirical Orthogonal Functions (EOF's) in space, and their Principal Components (PC's) in time, with the equation

$$NO(\Lambda, h, t) = \overline{NO}(\Lambda, h) + EOF_1(\Lambda, h) \cdot f_1(Kp) + EOF_2(\Lambda, h) \cdot f_2(\delta) + EOF_3(\Lambda, h) \cdot f_3(log(F10.7)), \qquad (1)$$

in which $h$, $t$, $Kp$, $\delta$, and $F10.7$ denote the altitude, time, planetary K-index, the solar declination, and the F10.7cm flux, respectively. However, $Kp$, $\delta$, and $F10.7$ may be insufficient to describe the processes responsible for the creation and depletion of NO in the MLT region.

Figure 6 demonstrates a situation in which the Kp-index may not suffice to model the NO number density near the magnetic pole. We use SOFIE measurements in this context since contrary to Odin SMR, SOFIE measures on consecutive days. Figures 6a) and 6b) compare SOFIE NO with the Kp-index at 102 km and -75°Λ during the Antarctic winter in 2013 and Antarctic summer in 2012 respectively.

Figures 6a) and 6b) display two key differences. Firstly, during summertime the NO number density is one order of magnitude lower than during winter. Additionally, Figure 6b) suggests that SOFIE NO varies with little or no lag with respect to fluctuations in the Kp-index. However, NO in Figure 6a) seems to have a lag of several days in comparison to the Kp-index. For instance, the Kp-index increases between days 142 and 146, while SOFIE NO follows an increase between days 146 and 149. Other examples can be seen between the increase in the Kp-index on days 151 and 157, as well as the corresponding increases

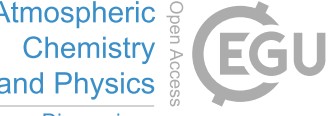

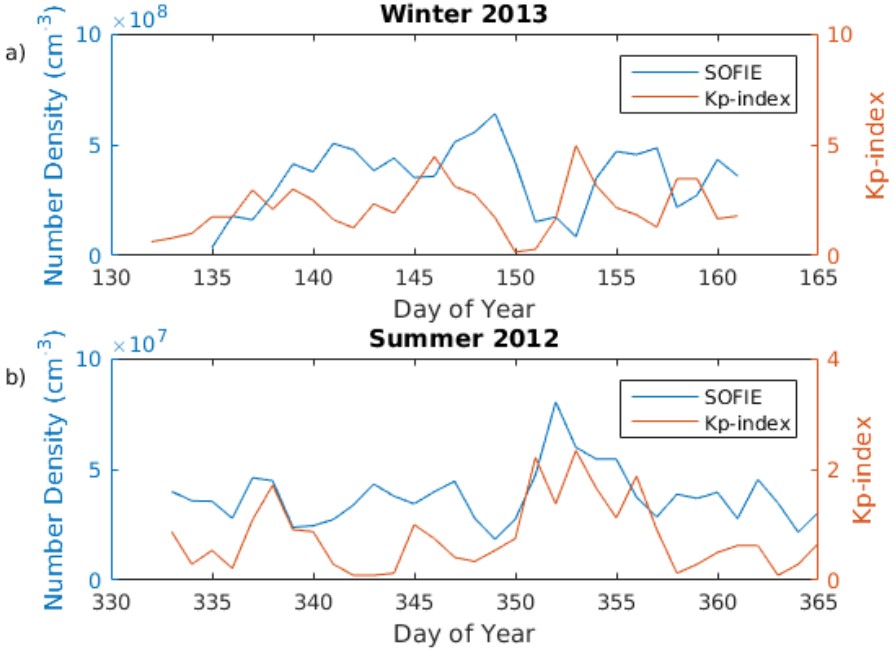

**Figure 6.** NO number density measured with SOFIE and Kp-index over time for a) 101 km altitude, 75° S magnetic latitude, winter 2013, and b) 101 km altitude, 75° S magnetic latitude, summer 2012.

in NO on days 153 and 158. Such a lag could be due to the longer lifetime of NO in dark conditions. Although Figure 6 offers no conclusive evidence of a major difference in the behavior of measured NO with respect to the Kp-index, it does warrant a need for further investigation. Perhaps the fact that NOEM disregards any effect of the Kp-indices of previous has an effect on the accuracy of the final model. This is why we propose two additional indices which take into account the Kp-index on prior

5  days as well. These are defined as:

$$com_1 = Kp_{d=1} + \sum_{d=2}^{n} Kp_d \cdot e^{-d/4} \cdot \frac{\delta + 23.4}{46.8}, \tag{2}$$

$$com_2 = Kp_{d=1} + \sum_{d=2}^{n} Kp_d \cdot e^{-d/4} \cdot sin(\frac{\delta * \pi}{46.8}), \tag{3}$$

in which $Kp_d$ is the Kp-index on day $d$, which is the number of days prior to the simulated day, $e$ is Euler's number, and n is a constant which depends on the solar declination. If the magnitude of the solar declination is less than 10°, 15°, 20°, or

10  more than 20°, n is assigned the value 2, 4, 6, or 11 respectively. A solar declination of 23.5° corresponds to summer solstice in the northern hemisphere. The two new indices aim to account for NO which has been created on previous days, since in dark conditions, NO in the MLT can persist for up to 10 days. The values in Equations 2 and 3 were chosen through iteration such





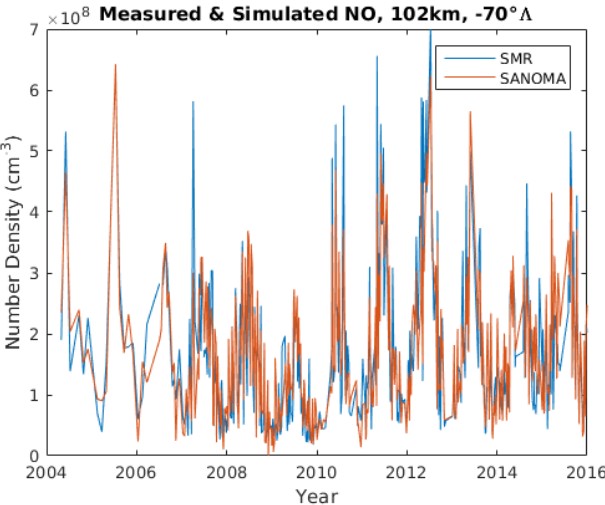

**Figure 7.** SMR-measured and SANOMA-simulated time series of NO at 102 km and $-70°\Lambda$

that SANOMA would explain the largest percentage of variance of the Odin SMR dataset. SANOMA uses the three indices from NOEM and the two indices defined in Equations 2 and 3, to build a multivariate linear fit.

The simulated NO for each altitude-latitude bin as a function of time becomes

$$NO(\Lambda, h, t) = K_p(t) \cdot c_1 + dec(t) \cdot c_2 + log(F10.7cm) \cdot c_3 + com_1(K_p, dec) \cdot c_4 + com_2(K_p, dec) \cdot c_5 + C \qquad (4)$$

5    in which the Kp-index and the logarithm of the F10.7cm flux both have a lag of 1 day for the best fit. This lag is in accord with previous similar studies (Hendrickx et al., 2015; Marsh et al., 2004; Solomon et al., 1999). Solomon et al. (1999) attribute this lag to the one-day lifetime of NO. Taking the logarithm of the F10.7cm flux instead of the flux itself results in a better fit between SANOMA and Odin SMR. This observation agrees with the findings of Marsh et al. (2004) and Fuller-Rowell (1993). The indices $c$ are obtained as an output of a multivariate linear fit function called 'regress' in MatLab 2015b, in which a matrix

10   containing all of the input indices as a function of time, and a corresponding matrix containing the measured NO time series in a given altitude and latitude bin, constitute the two inputs. Figure 7 shows the measured and simulated time series of NO for 102 km and -70°$\Lambda$. Although SANOMA generally follows the SMR measurements, it misses some of the highest peaks of NO. Either these peaks are results of random variation of the measurements, or SANOMA fails to simulate some physical or chemical process which would result in such peaks of NO. One such effect could be the dynamics of the MLT region, which

15   would transport NO.

The complete SANOMA model comprises of a total of 165 individual multivariate linear fits, one for each altitude-latitude bin. The coefficients of these fits indicate which of the input indices influence NO at the various locations. Figures 8a)-c) present the coefficients of the multivariate linear fits for the Kp-index, solar declination, and log(F10.7cm) flux, respectively. These figures offer an insight to where each of the physical drivers influence the NO number density the most. The effect of



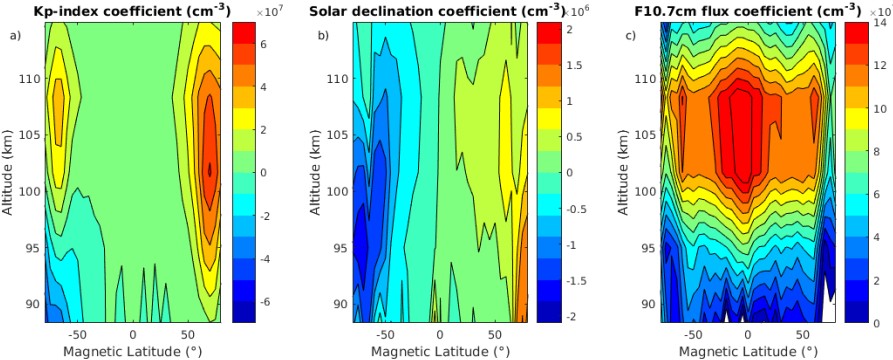

**Figure 8.** The coefficients of the a) Kp-index b) Solar declination c) F10.7cm flux, in Equation 4.

geomagnetic activity seems to be strongest around $\pm70°\Lambda$, while the solar radiation affects NO over the latitude range between $\pm70°\Lambda$ in the altitudes from 100 to 110 km. Figures 8a)-c) are also reminiscient of the EOF's obtained by Marsh et al. (2004). Furthermore, Figure 8b) shows the anti-symmetric effect of season on either contributing to an increase, or decrease of NO, as the coefficient changes sign from negative in the southern hemisphere, to positive in the northern hemisphere.

## 3.2 Comparing SANOMA to Odin SMR-measured NO

This section compares NO simulated with SANOMA with the original SMR-measured NO to confirm that the model has been succesfully built. SANOMA has a resolution of 6.66 km in altitude and $5°\Lambda$ in latitude. Figure 9 illustrates the mean of the difference of the time series of SMR and SANOMA NO at each location in $cm^{-3}$. The largest mean difference is $-2\times10^{-7}$ $cm^{-3}$, which means that on average, SANOMA can simulate the Odin SMR NO measurements with relatively low systematic error.

To explore the added value of the two new indices and the SANOMA equation, Figure 10 illustrates the R-squared values of three empirical models based on Odin SMR NO measurements. These models are a) a model built up according to Equation 1, called SMRNOEM, as a function of magnetic latitude and altitude, b) a model built with Equation 4 but without the compound indices, and c) with Equation 4. The R-squared value represents the percentage of the variance in the original SMR dataset which each model can explain. From left to right, we can observe an increasing trend in which the mean explained variances in each subplot is 0.509, 0.552, and 0.639. Figure 10b) demonstrates that using a multi-variate linear method instead of EOF analysis increases the R-squared somewhat in comparison to 10a), but the more obvious improvement is provided by the inclusion of Kp-index information from earlier days, as illustrated by Figure 10c). The most striking difference between Figures 10b) and c) is the great increase in the explained variance close to the magnetic poles and lower down in altitude, brought by the compound indices $com_1$ and $com_2$. This improvement can be explained with by fact that these two indices take the lifetime of upper-atmospheric NO into account, hence yielding more accurate results in the polar regions in which the accumulation of NO in the absence of sunlight plays an important role. Using Equation 4 to build an empirical model therefore increases the overall




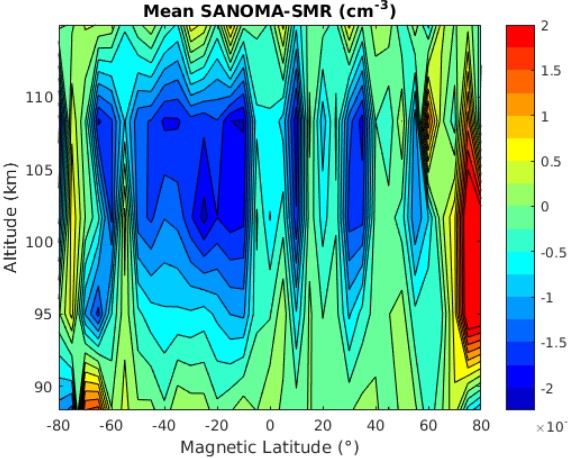

**Figure 9.** Mean of the differences between the time series of SANOMA-simulated and SMR-measured NO in cm$^{-3}$.

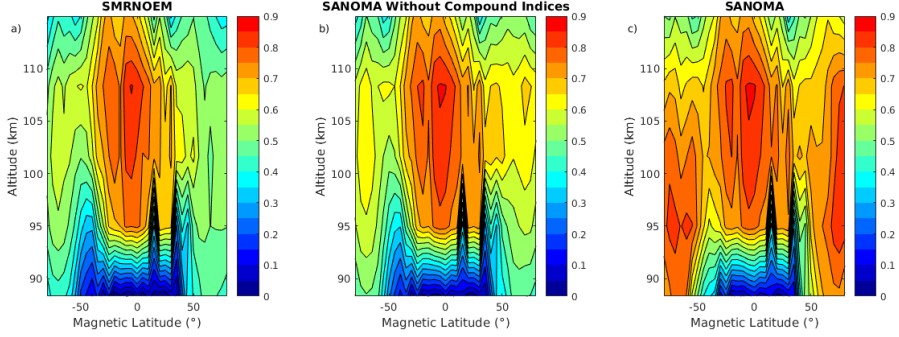

**Figure 10.** R-squared between SMR and an empirical model from Odin SMR NO using a) Equation 1 b) Equation 4 without $com1$ or $com2$ c) Equation 4.

explained variance by 13%, and by up to 40% near the poles. The figures imply, that Equation 4 would describe the variation of NO best around the magnetic poles, in which geomagnetic activity dominates its variation. Although the R-squared attains 0.8 close to the magnetic poles and exceeds 0.5 in most areas, values down to 0.1 indicate that SANOMA fails to accurately simulate the variability of NO at magnetic latitudes closer to the equator than $\pm 40°$ and below altitudes of roughly 95 km.

5  So far we have presented how well SANOMA explains SMR measurements. However, the SMR measurements themselves include measurement error and hence a model will be unable to perfectly reproduce the measurements. We can attempt to separate the discrepancy between SANOMA and SMR into two parts: the measurement error from Odin SMR and the modeling



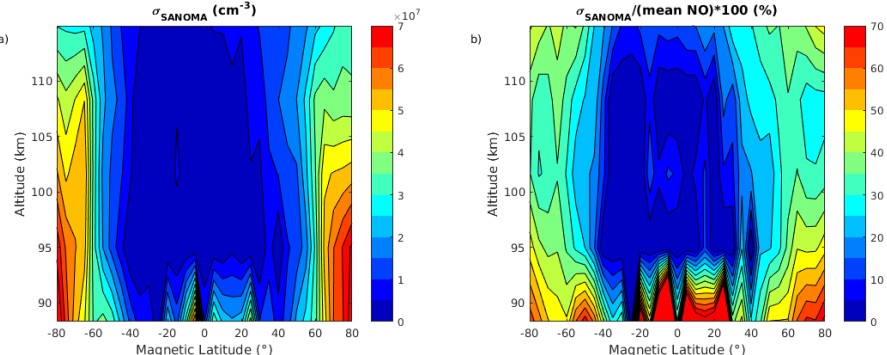

**Figure 11.** a) Standard deviation of the modeling error b) Standard deviation of the modeling error relative to the mean NO.

error from SANOMA. Having an understanding of the error of SANOMA can help to assess the reliability of its simulations. Assuming that all errors are normally distributed, we estimate the variance of the modeling error with

$$\sigma^2_{SANOMA} = \sigma^2_{total} - \sigma^2_{OdinSMR}, \tag{5}$$

in which $\sigma^2_{SANOMA}$, $\sigma^2_{total}$, and $\sigma^2_{OdinSMR}$ are the estimated variances of the model error, total error, and measurement error.

For each location, $\sigma^2_{total}$ is estimated as the variance of the time series of SANOMA-SMR. The mean of the daily means of the known measurement error constitutes $\sigma^2_{OdinSMR}$. In some cases, $\sigma^2_{SANOMA}$ is close to zero, or slightly negative. If this occurs, $\sigma^2_{SANOMA}$ is set to zero, so we can calculate the square root of the variance to yield the standard deviation of the modeling error. Figure 11 illustrates the standard deviation of the modeling error in absolute and relative values. Figure 11b) suggests that the highest relative modeling error is located at the lowest altitudes. One explanation could be that dynamics start

to play a larger role lower down, and since SANOMA includes no dynamics, it fails to model the variations of NO accurately. This could also be due to a lower signal-to-noise ratio in the measurements at lower altitudes.

## 4   Assessment of SANOMA and NOEM

So far, we have presented SANOMA, its underlying principle, and a comparison of its simulations with NO measured by Odin SMR. This section evaluates the performance of SANOMA and NOEM by comparing simulated NO number density

with measured NO from the four independent instruments, SOFIE, SCIAMACHY, ACE-FTS, and MIPAS. SANOMA can be seen as a tool to compare the SMR dataset with the other data, and therefore these comparisons can provide valuable information regarding the accuracy of SMR-NO measurements. For an overview, Table 1 lists the underlying techniques behind the instruments as well as the time and regions of operation.

  SANOMA has a resolution of 6.66 km in altitude and 5°Λ in latitude while NOEM has corresponding resolutions of 3.33 km

and 5°Λ. For direct comparison, we average two adjacent NOEM altitude bins to result in an altitude resolution of 6.66 km.



**Table 1.** Overview of all the compared NO datasets

| Instrument | Period | Latitudes | Altitudes (km) | Technique | Wavelength |
|---|---|---|---|---|---|
| SMR | 2003- | 82° S - 82° N | 85-115 | Microwave Emission | 544 $\mu$m |
| SNOE | 1998-2000 | 82° S - 82° N | 100-150 | UV Scattering | 215 nm, 237 nm |
| SOFIE | 2007- | 80° S - 60° S, 60° N - 80° N | 40-140 | Solar Occultation | 5.316 $\mu$m |
| SCIAMACHY | 2008-2012 | 88.75° S - 88.75° | 60-160 | UV Scattering | 230-314 nm |
| ACE | 2004- | 82° S - 82° N | 70-110 | Solar Occultation | 5.18-5.43 $\mu$m |
| MIPAS | 2005-2012 | 82° S - 82° N | 70-120 | Infrared emission | 5.3 $\mu$m |

To prepare each data set comparison with these models, an algorithm sorts the NO measurements from each instrument into the same bins as in SANOMA and calculates daily average NO number density. The following four subsections present the comparisons of SANOMA and NOEM to the four independent instruments, while Section 4.5 summarizes these results.

## 4.1 SOFIE

This section compares the SANOMA and NOEM simulated NO number density with Level 2 V1.3 SOFIE measurements (Gordley et al., 2009) (available at sofie.gats-inc.com) between 102 km to 115 km. These altitudes reflect the range in which NOEM, SANOMA, and SOFIE all overlap. According to Figure 12, especially prior to 2011, NOEM-simulated NO exceeds that of both SOFIE-measured and SANOMA-simulated NO. A possible cause for this effect is discussed in Section 4.5. Additionally, the SOFIE measurements show high peaks of NO, which are partially absent in SANOMA, and completely

absent in NOEM. The SANOMA time series seems to generally agree better with the SOFIE measurements than NOEM.

    To examine the accuracy of SANOMA and NOEM as a function of magnetic latitude and altitude, Figure 13 depicts the median of the difference between the simulated NO and SOFIE as a percentage of the mean SOFIE number density. Although SOFIE predominantly measures at geographic latitudes between 60 and 80 degrees in both hemispheres, some measurements from the southern hemisphere originate from as low as 40°Λ due to a changing orbit and greater offset between the geomagnetic

and geographic poles in the southern hemisphere. Generally, the difference between SANOMA and SOFIE lies between ± 40% in the region from 60°Λ to 80°Λ in both hemispheres while the NOEM-modeled NO exceeds SOFIE with up to 160%.

    To assess the amount of variation that each model captures of the original SOFIE data, Figure 14 presents the R-squared values of linear fits between SANOMA and SOFIE as well as NOEM and SOFIE, respectively. Whereas SANOMA exhibits up to 0.65 R-squared at -70° Λ and 102 km, NOEM falls significantly shorter with a maximum R-squared value of 0.35 at

-55°Λ and 102 km. Furthermore, Figure 14 emphasizes that SANOMA outperforms NOEM at all altitudes in the auroral zone of 60°Λ to 80°Λ in both hemispheres and that SANOMA captures more variability of the SOFIE measurements in the southern than in the northern hemisphere.





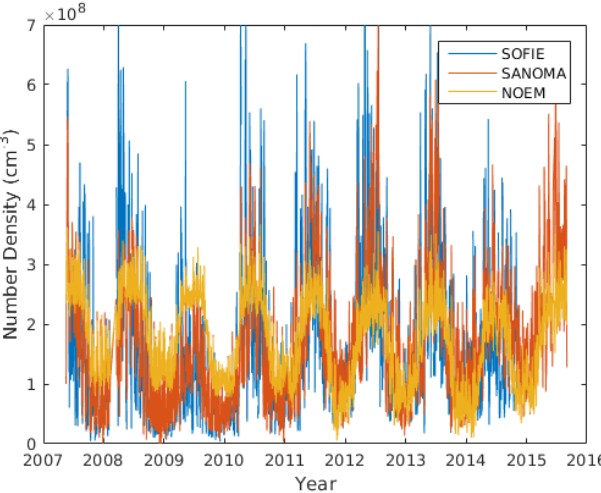

**Figure 12.** Time series of NO number density measured with SOFIE over the entire measurement period as well as simulated NO with SANOMA and NOEM, 102 km altitude, -70°Λ.

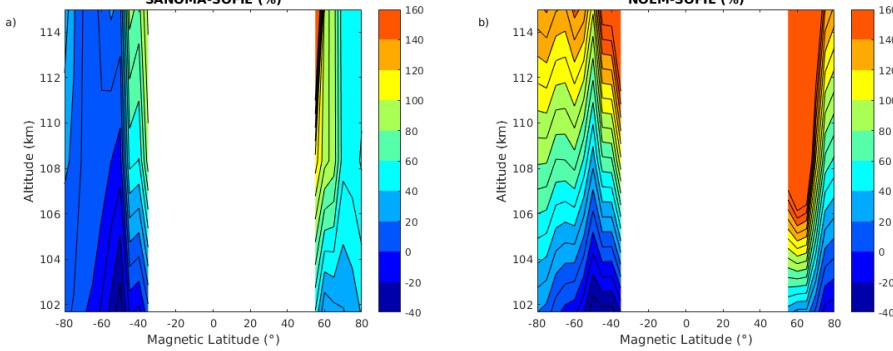

**Figure 13.** a) Median difference between SANOMA and SOFIE as a percentage of mean SOFIE NO number density. b) Median difference between NOEM and SOFIE as a percentage of mean SOFIE NO number density.

## 4.2 SCIAMACHY

This section compares the SANOMA- and NOEM-simulated NO number density with v1.8.1 SCIAMACHY (Bender et al., 2013) measurements of NO in the altitudes from 102 km to 115 km. Figure 15 draws the time series of SCIAMACHY-measured, as well as SANOMA- and NOEM-simulated NO for 102 km and -70°Λ. As in Figure 12, NOEM NO exceeds that of both SANOMA and SCIAMACHY whereas SANOMA follows SCIAMACHY measured NO more accurately over the entire measurement period. Another clear feature is that SANOMA NO increases significantly more than SCIAMACHY NO during Antarctic winter. One possible explanation is that SCIAMACHY only measures day-time NO, whereas the Odin SMR data





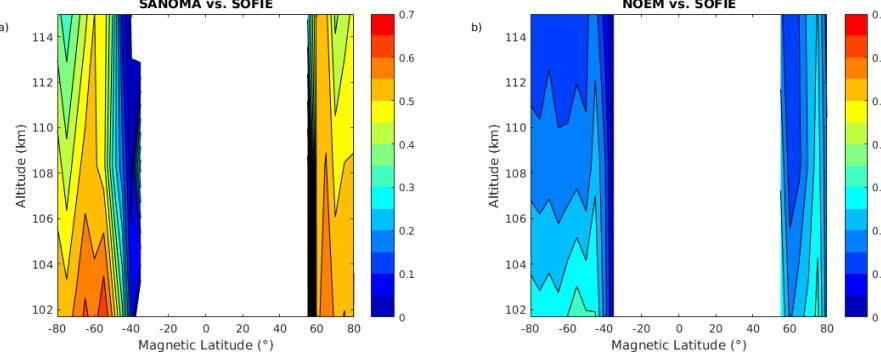

**Figure 14.** a) Adjusted R-squared of linear fits between SANOMA and SOFIE NO number density. b) Adjusted R-squared of linear fits between NOEM and SOFIE NO number density.

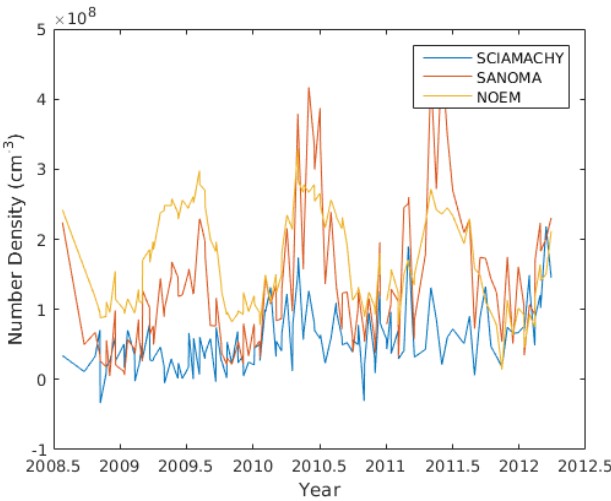

**Figure 15.** Time series of NO number density measured with SCIAMACHY and simulated with SANOMA as well as NOEM, 102 km altitude, -70° Λ.

which resulted in SANOMA also included measurements from the polar night, during which the NO number density is enhanced.

Figures 16a) and b) display the mean difference between SANOMA and SCIAMACHY as well as NOEM and SCIAMACHY as a percentage of the mean SCIAMACHY number density. NOEM NO exceeds SCIAMACHY with over 200%
5  while SANOMA reaches maximum differences up to 150% at -70° Λ and 102 km.

As can be seen in Figures 17a) and b), SANOMA reaches maximum R-squared values of 0.5 in the areas of high auroral activity while NOEM attains only 0.25 at 102 km and -60° Λ. The results suggest that SANOMA captures significantly more variability of the SCIAMACHY dataset than NOEM.



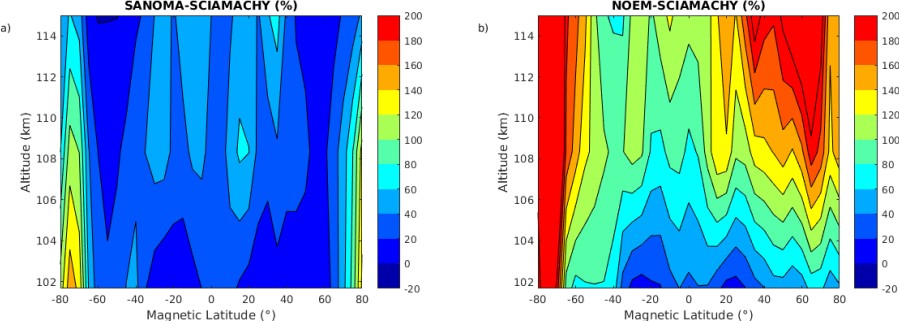

**Figure 16.** a) Median difference between SANOMA and SCIAMACHY as a percentage of mean SCIAMACHY NO number density. b) Median difference between NOEM and SCIAMACHY as a percentage of mean SCIAMACHY NO number density.

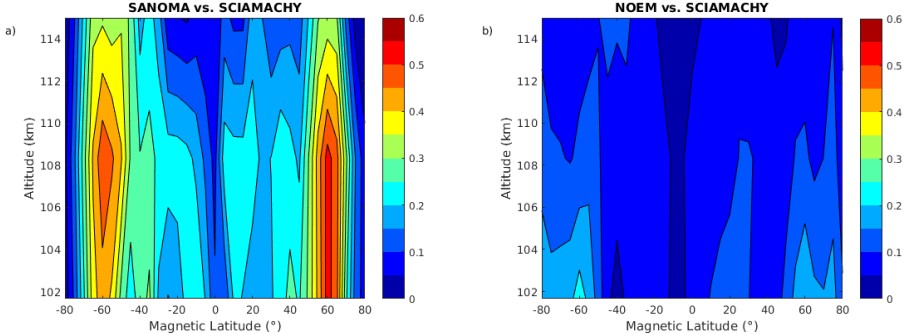

**Figure 17.** a) Adjusted R-squared of a linear fit between SANOMA and SCIAMACHY NO number density. b) Adjusted R-squared of a linear fit between NOEM and SCIAMACHY NO number density.

### 4.3 ACE

This section compares the SANOMA and NOEM simulated NO number densities with V3.5 ACE measurements (Bernath et al., 2005) between 102 and 108 km. Quality filtering of the ACE NO measurements constitutes the reason for the upper limit of the altitude in these comparisons. Figure 18 provides an overview of almost one entire solar cycle of ACE, SANOMA, and
5  NOEM NO. It can be seen that both models miss the highest spikes of NO in the ACE measurements, although SANOMA manages to capture several between 2005 and 2012. Figure 19a) suggests that the difference between SANOMA and ACE is generally zero or positive above 105 km with a maximum difference of 60% within 30° Λ of the magnetic equator at 108 km. Negative differences can be observed below 105 km, with a minimum of -50% near the magnetic equator at 102 km. On the contrary, Figure 19b) shows that the difference between NOEM and ACE is positive at nearly all locations, with a maximum of
10  160% at 20° N magnetic latitude at 108 km. The patterns in Figure 19 could be due to the different height at which each model places the maximum NO. The maximum of NO number density in ACE is located close to 100 km, while the corresponding SANOMA and NOEM NO maxima are located between 102-108 km.





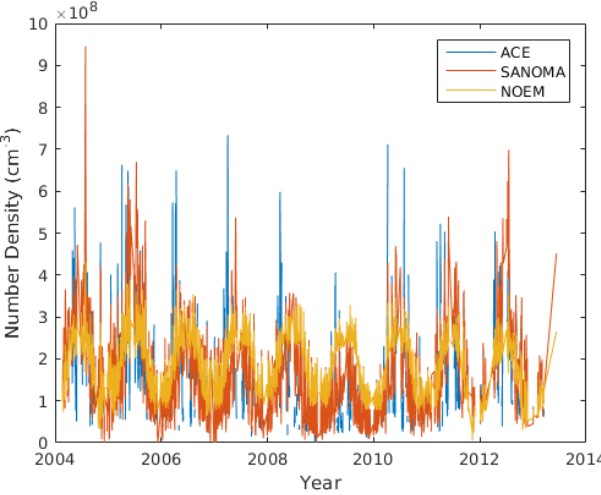

**Figure 18.** Time series of NO number density measured with ACE and simulated with SANOMA as well as NOEM, 102 km altitude, -70° Λ.

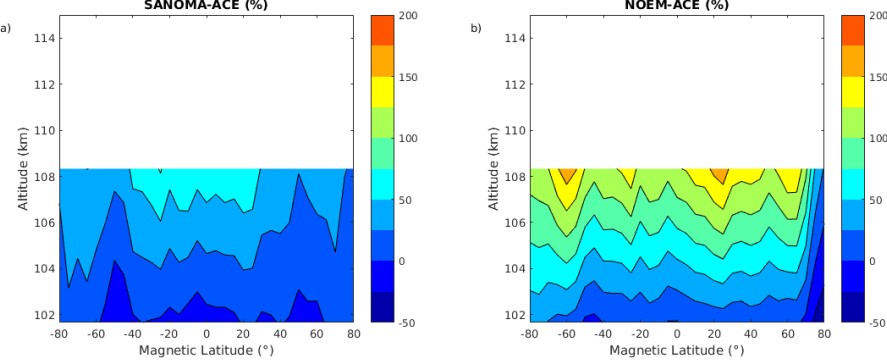

**Figure 19.** a) Median difference between SANOMA and ACE as a percentage of mean ACE NO number density. b) Median difference between NOEM and ACE as a percentage of mean ACE NO number density.

The SANOMA R-squared values in Figure 20a) reach a maximum of 0.65 at 70° Λ magnetic latitude while Figure 20b) indicates that NOEM only attains 0.35 at 35° Λ at 102 km. Figure 20a) also shows a band of lower R-squared from -45° Λ to -25° Λ, the cause of which remains unknown.

## 4.4 MIPAS

5 Finally, this section compares the simulated NO number densities with v5r MIPAS 620 (Fischer et al., 2008) measured NO in the altitudes from 102 km to 115 km. In Figure 21, although both NOEM and SANOMA seem to follow the variations of MIPAS-measured NO, SANOMA more accurately reproduces the number density of MIPAS. Figures 22a) and 22b) display





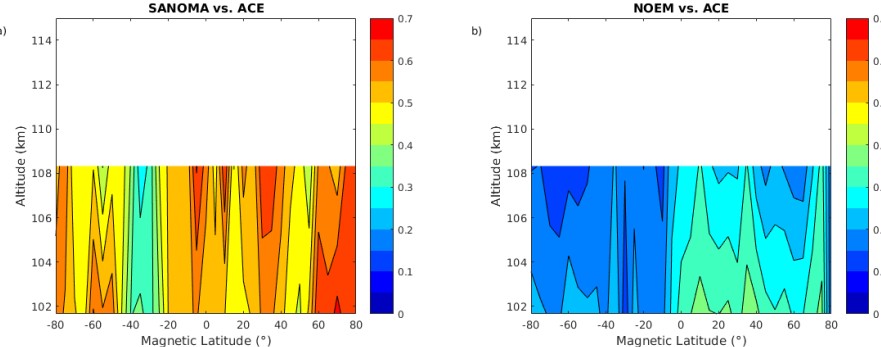

**Figure 20.** a) Adjusted R-squared of a linear fit between SANOMA and ACE NO number density. b) Adjusted R-squared of a linear fit between NOEM and ACE NO number density.

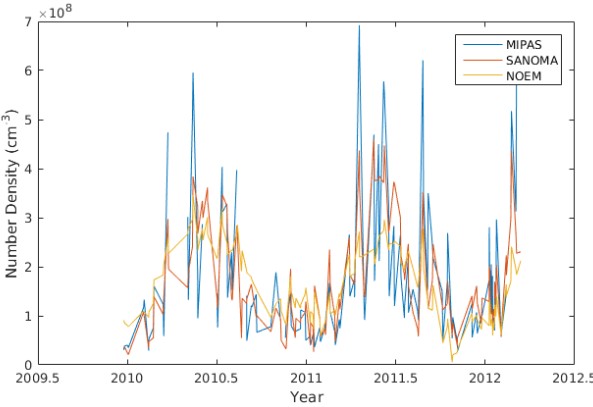

**Figure 21.** Time series of NO number density measured with MIPAS and simulated with SANOMA as well as NOEM, 102 km altitude, -70° Λ.

the median differences between SANOMA and MIPAS as well as NOEM and MIPAS as a percentage of the MIPAS-measured NO. SANOMA underestimates MIPAS NO at all locations, with a minimum of -60% at 115 km around 70° Λ. Meanwhile, the difference between NOEM and MIPAS changes from positive in the higher altitudes to less positive or negative in the lower altitudes. Finally, Figures 23a) and 23b) suggest that in both models, the R-squared varies strongly with latitude. SANOMA explains up to 80% of the variance of MIPAS NO at the lower altitudes around 70°N, while NOEM only reaches 50% at 20°N.

## 4.5 Summary of the Results

The section aims to summarize and elaborate on the results of the previous sections. To achieve an overview of the results, Table 2 presents R-squared values between the measurement instruments and three different models: a) NOEM b) SMRNOEM, a model built using the NOEM equations, but Odin SMR data, and c) SANOMA. Comparing NOEM and SMRNOEM informs



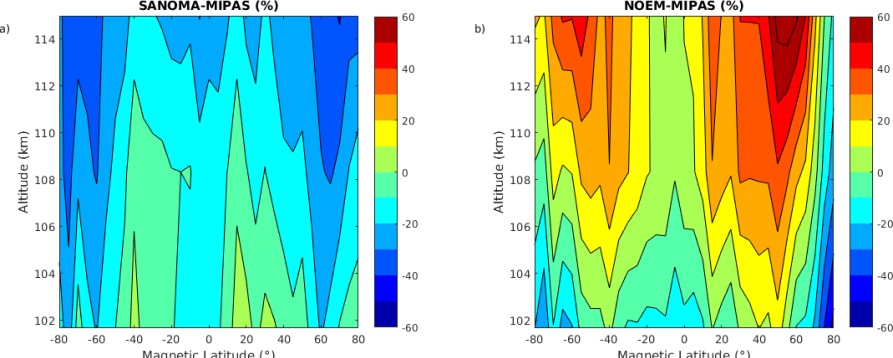

**Figure 22.** a) Median difference between SANOMA and MIPAS as a percentage of mean MIPAS NO number density. b) Median difference between NOEM and MIPAS as a percentage of mean MIPAS NO number density.

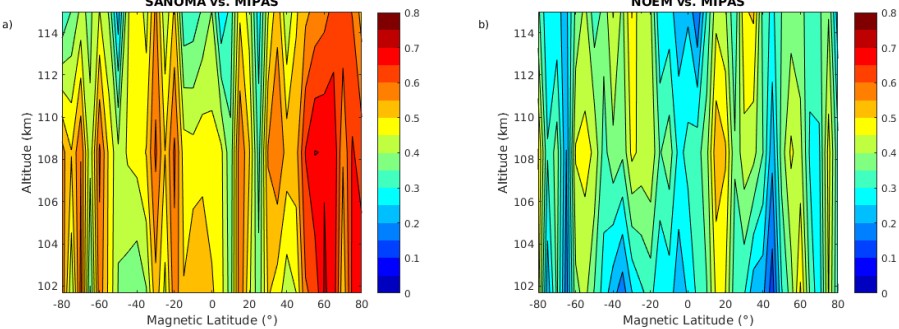

**Figure 23.** a) Adjusted R-squared of a linear fit between SANOMA and MIPAS NO number density. b) Adjusted R-squared of a linear fit between NOEM and MIPAS NO number density.

us on any improvement provided by using the Odin SMR NO dataset instead of the SNOE dataset, while differences between SMRNOEM and SANOMA are results of the different approach to developing the two models.

Table 2 reveals that SANOMA consistently explains a similar amount, or more variance than SMRNOEM in all regions and for all instruments, which in turn explains more variance than the SNOE-based NOEM. With SOFIE, ACE, and SCIAMACHY,

5  the median difference between SANOMA and these instruments is closer to zero than for NOEM, while the opposite holds true for MIPAS. Generally, the models capture more variance in the northern, than in the southern hemisphere. Perhaps the larger offset between the geomagnetic and geographic pole, or the more stable polar vortex in the souther hemisphere affect the amount of NO in ways which are beyond the reach of these simple models.

The SMRNOEM values in Table 2 highlight how much the Odin SMR dataset itself improves the resulting empirical model

10  and the consistently higher R-squared values in SANOMA compared to SMRNOEM justify the use of the compound indices. Using the SANOMA- instead of the NOEM-equation with SMR-measured NO increased the percentage of explained variance



**Table 2.** Mean R-squared between 102 and 115 km in the southern high latitudes, tropics, and northern high latitudes as well as the mean of the mean percent difference over these three domains between the various instruments and a) NOEM, b) SMRNOEM, and c) SANOMA.

| Instrument | R-sq (-75°Λ to -55°Λ) | | | R-sq (-30°Λ to 30°Λ) | | | R-sq (55°Λ to 75°Λ) | | | Mean Difference (%) | | |
|---|---|---|---|---|---|---|---|---|---|---|---|---|
| | a | b | c | a | b | c | a | b | c | a | b | c |
| SMR | 0.287 | 0.509 | 0.645 | 0.544 | 0.704 | 0.730 | 0.356 | 0.487 | 0.696 | 42.8 | 2.2 | 0.1 |
| SOFIE | 0.193 | 0.307 | 0.495 | N/A | N/A | N/A | 0.212 | 0.207 | 0.424 | 112.9 | 38.6 | 38.7 |
| SCIAMACHY | 0.124 | 0.292 | 0.309 | 0.069 | 0.148 | 0.159 | 0.118 | 0.095 | 0.316 | 117.7 | 53.6 | 37.1 |
| ACE | 0.169 | 0.423 | 0.507 | 0.246 | 0.469 | 0.509 | 0.272 | 0.381 | 0.572 | 69.1 | 33.8 | 26.7 |
| MIPAS | 0.325 | 0.483 | 0.470 | 0.350 | 0.456 | 0.459 | 0.378 | 0.378 | 0.632 | 9.3 | -17.9 | -18.4 |

by up to 100% near the magnetic poles. Moreover, these comparisons stem from 102 km and upwards, but as indicated by Figure 10, SANOMA outperforms SMRNOEM by even more in the altitudes below 102 km.

Since SMR measures both day and night time NO, a positive difference compared to the daytime measuring instrument SCIAMACHY was expected, but SANOMA shows a positive difference in comparison to SOFIE and ACE as well. The mag-
nitude of the relative differences between SANOMA and each satellite is similar, although slightly higher than the differences between the SMR dataset and the other instruments described by (Bender et al., 2015). As the time series in Figures 12, 15, 18, and 21 showed, NOEM overestimates the measured NO prior to 2011. The fact that NOEM was derived from SNOE data between 1998-2000, a time of high solar activity, might explain why it fails to accurately reproduce the lower NO number densities observed at times of lower solar activity. SANOMA profits from the wider variety of solar conditions experienced
over the Odin SMR measurement period from 2004 to 2016 to provide more accurate NO number density over the entire solar cycle.

On average, NOEM NO seems to be more accurate than SANOMA only in comparison to MIPAS. However, the MIPAS measurements should be treated with care, since Bender et al. (2015) found that MIPAS NO is 80-120 % higher at altitudes between 100 and 120 km. Because the two empirical models can be regarded as extensions of the original measurement instru-
ments, a high MIPAS NO relative to SANOMA was expected.

# 5   Conclusion and Discussion

This study presented a new empirical model called SANOMA to simulate NO in the MLT. This model is based on V3.0 Odin SMR NO, to which we fit multivariate linear functions using the Kp-index, solar declination, the logarithm of the F10.7cm flux, as well as two compound indices based on the Kp-index and solar declination. These two compound indices attempt
to account for the lifetime of NO in the absence of sunlight. SANOMA can capture an average of 63.9% of the variance of the Odin SMR NO between 88 and 116 km and -80°Λ to 80°Λ. The comparisons of SANOMA, the model developed in this study, and NOEM, a similar model by Marsh et al. (2004), with measurements from the SOFIE, SCIAMACHY, ACE, and MIPAS instruments suggest that SANOMA explains significantly more variance of the NO measured by each instrument than





NOEM. The percentage of explained variance by SANOMA spans from a minimum of 16% in the magnetic tropics (-30°Λ to 30°Λ) with SCIAMACHY, to 73.2% in the magnetic tropics with SMR. Similarly, NOEM captures a minimum of 6.9% of the variance with SCIAMACHY NO in the magnetic tropics, and a maximum of 54.5% of the variance with SMR NO in the same region. Furthermore, the results suggest that over the entire measurement period in this study, NOEM overestimates the

amount of NO for all of the satellites, while SANOMA either over- or underestimates it. The difference between SANOMA and measured NO is closer to zero than NOEM for SOFIE, SCIAMACHY, and ACE, while the opposite holds true for MIPAS. This difference to MIPAS NO is consistent with the findings of Bender et al. (2015), who found that MIPAS NO is roughly 100% higher in the altitudes 100-120 km than in ACE, SMR, or SCIAMACHY.

An alternative to the multivariate linear fit in this study would have been EOF analysis such as in (Marsh et al., 2004). We

attempted this approach and compared the resulting model to SANOMA, with roughly equivalent success. The multivariate linear fit approach was then chosen for its simplicity.

Our original hypothesis that a model similar to NOEM, but derived using Odin SMR data, would result in a more accurate model was proven to be true. Comparing the results of NOEM and SANOMA with measured NO showed that especially during times of low solar activity, NOEM overestimates NO by roughly 100%. This could be attributed to the fact that NOEM was

built on only two years of SNOE NO data from 1998-2000, a period of high solar activity. Hence, when the model is applied to low-activity periods, such as 2009-2010, the extrapolation from high-activity to low-activity conditions is inaccurate, resulting in large errors of NOEM NO compared to the measurements.

In terms of explaining the variation of NO, unlike NOEM, SANOMA manages to recreate more of the highest concentrations of NO. SANOMA still fails to explain some of the highest spikes of NO and suffers from a relatively coarse (6.5 km) altitude

resolution as well as a narrow altitude range (85 km-115 km). The results from Figures 11 suggested that SANOMA fails to model some physical processes which govern the amount of NO. Perhaps dynamical processes cause fluctuations in the concentration of NO at 85 km, causing the model to miss some of the variation of NO. However, the error associated with SANOMA has been estimated and is available to any potential user of the model.

Creating SANOMA with all SMR-measurements will have likely introduced a positive bias compared to day-measuring

instruments, such as SCIAMACHY, since night-time NO is expected to be higher than day-time NO. An alternative to the current model would be to provide two versions of SANOMA: one for day, and one for night.

Although no rigorous validation of Odin SMR NO in the MLT regions exists, Bender et al. (2015) proposed that the Odin SMR is consistent with measurements from other satellites. However, it is conceivable that all of these measurements could deviate from the true concentration of NO. Even so, SANOMA offers an estimate which reflects the physical processes behind

the creation of NO. As long as no in-situ measurements are available, remote sensing is the only way to provide some estimate of the true state. SANOMA could be used in the future as an input for chemical models of the atmosphere, as apriori information for satellite retrievals of NO, or as a transfer function to compare NO observational data sets with each other, to name a few possible applications. SANOMA and accompanying scripts are available on odin.rss.chalmers.se

*Author contributions.* Joonas Kiviranta: main body of text, work behind SANOMA, analyzed satellite and indices data, all plots Kristell Pérot: initiated the study, scientific consulting, editing of text, input on correctness of facts, etc. Donal Murtagh, scientific consulting, editing of text, Patrick Eriksson, scientific consulting, editing of text.

*Acknowledgements.* Odin is a Swedish-lead satellite project funded jointly by Sweden (SNSB), Canada (CSA), Finland (TEKES), France (CNES) nad the Third-Party Missions programme of the European Space Agency (ESA). The following people have kindly provided their support by the method indicated in the brackets: Koen Hendrickx (providing SOFIE data), Daniel Marsh (providing NOEM and general feedback), Bernd Funke (providing MIPAS data), Kaley Walker (providing ACE data), Stefan Bender (providing SCIAMACHY data), Jean Lilensten (providing general feedback on the model).



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

untitled.