# Peer review of "An empirical model of nitric oxide in the upper mesosphere and lower thermosphere based on 12 years of Odin-SMR measurements"

_Atmospheric Chemistry and Physics, 2018_

## Referee Comment (RC1) · Anonymous Referee #1 · 2 Mar 2018

This paper uses Odin SMR NO observations taken between 2004 and 2016 in the upper mesosphere and lower thermosphere to build an empirical model of NO number densities in the altitude range 85 -115 km. The model is compared to the previous NOEM empirical model which builds on SNOE NO observations taken between 1998 and 2000,

The presented empirical model is very useful for constraining or validating atmospheric models and improves upon previous empirical models by considering a longer observational period that covers the entire solar cycle. It is also very useful as transfer function allowing for indirect intercomparison of thermospheric NO observations with disparate

temporal sampling. Such indirect intercomparisons of the SANOMA model (and hence SMR) to satellite observations from ACE, MIPAS, SCIAMACHY and SOFIE are also presented.

The paper is well written and I have only minor comments specified below:

p2 l26: "Additionally, the amount of NO influences the thermal balance of the MLT, especially at ultra-vialett wavelengths (Richards et al., 1982)". The cited paper discusses the role of NO in modulating the thermospheric heating efficiency by infrared 5.3 um cooling (not at ultra-violet wavelengths).

p3 l25: "since night-time NO is expected to be higher than day-time NO". It should be mentioned that the day-night ratio of NO depends on latitude and altitude (see, e.g. Bermejo-Pantaleon et al., 2011)

p3 l27: typo: based -> based on

p4 l25: "SANOMA will express NO in number density to accommodate chemical models" Why is number density over altitude better suited to accommodate chemical models than vmr over pressure level?

p8 l 6: the PC's mentioned here are not included in Eq. 1. It would be clearer to note that the PCs were replaced by polynomial fits to the geophysical parameters Kp, delta, log(F10.7) in Marsh et al., 2004, represented in Eq 1 by f1,f2, and f3.

p10 l9: Have auto-correlations been considered in the regression? This would be important since explained variances might be overestimated otherwise (see, e.g. Hendrickx et al., GRL, 44, 2017)

p 11 l11: "R-sqared" coefficient of determination

p11 l8: Why aren't the mean differences between SANOMA and SMR zero in a given bin? This would be expected from a simple multilinear regression.

p13 l10: "SANOMA includes no dynamics". Since SANOMA is an empirical model it

does not include any physical process.

p13 l11: "This could also be due to a lower signal-to-noise ratio in the measurements at lower altitudes." Isn't the measurement error already accounted for in Eq 5?

p14 Table 1: Isn't ACE measuring at altitudes < 70 km as well?

p14 Table 1: Same for MIPAS. Also, the MIPAS latitude range should be 90S - 90N.

p 15 Fig 12: I'm puzzled about the NOEM results, showing minimum polar summer NO concentrations in 2012-2014, i.e., well after the solar cycle minimum (2009-2010). How is this possible if NOEM uses F10.7 and Kp (both having lower values in solar min)?

p18 l5 : The given reference is correct for MIPAS but for the NO data version mentioned here Bermejo-Pantaleon et al, 2011 would be more adequate. It is unfortunate that this study uses data version v5r 620 since it only covers the years 2010-2012. The newer version v5r 622 (available, for instance, at http://mesospheo.fmi.fi/data_service.html) covers the entire period 2005-2012.

p19 l2: "with a minimum of -60% at 115 km around 70 deg". It looks more like a minimum of -30 to -40% at 115 km in Fig. 22.

p21 Table 2: caption: "the mean of the mean percent difference". Do you mean the mean percent difference averaged over the respective latitude bands? Further: "between the various instruments and a) NOEM, b) SMRNOEM, and c) SANOMA". Shouldn't it be the other way round, i.e., model - instrument?

p21 l4: Since SMR measures both day and night time NO, a positive difference compared to the daytime measuring instrument SCIAMACHY was expected, but SANOMA shows a positive difference in comparison to SOFIE and ACE as well". SOFIE and ACE are solar occultation instruments. A diurnal sampling bias can therefore no be excluded when comparing to instruments that measure at both day and nighttime (like SMR or MIPAS).

p21 l12-15: I wouldn't say that MIPAS measurements should be treated with care, particularly because MIPAS shows small absolute difference with respect to SANOMA (and hence SMR), as well as high R-squared values (Table 1). It is likely that the apparently different behavior of MIPAS compared to other datasets is related to the shorth time coverage of the MIPAS data version used here (2010-2012). It would be very interesting to check this by using the newer version v5r 622 (2005-2012).

p22 l5-7: see above.

---

## Referee Comment (RC2) · Anonymous Referee #2 · 4 Mar 2018

The authors present a new model of lower thermospheric NO, based on observations by the SMR instrument. The purpose is to provide the scientific community with a new, improved proxy for thermosphere NO and the authors throughly compare their regression model to most available observations since 1998 (for some reason HALOE has been left out of the comparison). The model presented in the paper is potentially very useful for thermosphere upper boundary conditions in high top models, but I have concerns on the presentation of how the model was built which I will explain below. In my opinion these clarifications are vital for others to be able to 1) evaluate and 2) apply this model, and thus I am recommending a major revision so the authors can clarify these points. Clearly the model compares nicely with independent observations, but

these clarifications are still needed. If we are to use the model (and I think many people will be interested!), we need to know what went in. Particularly as the approach is so different from other linear regression models in the literature. My comments below may sound critical, but remember that the purpose is to clarify your work so that we can use it with confidence.

How was is the regression model built? We have the equation:

NO(lamda, h, t) = Kp(t)*c1 + dec(t)*c2 + log(F10.7(t))*c3 + com1(Kp(t),dec(t))*c4 + com2(Kp(t),dec(t))*c5 + C

Firstly it is not clear how the authors formed com1 and com2. The paper says that they were a result of iteration, but there is no physical explanation for them or clear explanation of the iteration process. They do, however, introduce autocorrelation with all except term #3. I am worried that there are two terms which both have a linear (lagged) Kp term as well as linear/sin terms depending on solar declination in addition to the linear terms in #1 and #2 - without any explanation.

The final number of terms in #4 and #5 also depend on solar declination in a way that is not explained clearly. Why for summer solstice conditions are there a maximum number of lagged days (11)? Would it not be natural to assume a lag is needed for winter conditions when the lifetime of NO is larger and thus there might be build-up? Why was the Kp lag not built into the first term of the regression (and same for solar declination)? What is the physical meaning of having these extra Kp & dec dependent terms?

The actual regression coefficients "c1, c2,...c5" are not given (First 3 were plotted in Figure 8, I didn't notice remarks on c4 and c5) and thus the equation can not be used. Please note that many readers will not have access to Matlab statistics toolbox used for the analysis.

Other comments: Figure 2: This is plot of the F10.7 time series, clearly taken directly

from the NOAA website. Although the reference to the website is given, please plot the data yourself. This is a very simple figure with the actual monthly means and a running mean. The future prediction (red line) is not necessary for this paper.

The indices you use here are not "space weather" indices, but rather geomagnetic and solar indices. The term space weather has a very specific meaning relating to impacts on technology (and these indices can be used for those), but as we are looking at impacts on the Earth's atmosphere we should talk about geomagnetic and solar indices.

The AE index (Auroral Electrojet) is mentioned in the same section, but this is not used anywhere after that, only Kp is mentioned.

---

## Author Comment (AC1) · 9 Jun 2018

We would like to thank the two anonymous referees and the editor for their attention to our paper "An empirical model of nitric oxide on 12 years of Odin-SMR measurements". The comments were constructive and helped us to significantly improve the paper. Below, the reviewers' comments are marked with (1), the author's response with (2), and the change in the manuscript with (3).

The changes to the manuscript can be viewed in the pdf attached as a supplement to this response, where changes have been marked with difflatex.

[Figure]

Reviewer 1 comments:

(1) p2 l26: "Additionally, the amount of NO influences the thermal balance of the MLT, especially at ultra-vialett wavelengths (Richards et al., 1982)". The cited paper discusses the role of NO in modulating the thermospheric heating efficiency by infrared 5.3 um cooling (not at ultra-violet wavelengths).

(2) The reviewer correctly points out a mistake in the content of the sentence.

(3) Changed sentence to refer to infrared cooling.

(1) p3 l25: "since night-time NO is expected to be higher than day-time NO". It should be mentioned that the day-night ratio of NO depends on latitude and altitude (see, e.g. Bermejo-Pantaleon et al., 2011)

(2) The original manuscript was inaccurate in its description of the day/night NO differences, although in reality the relationship is more nuanced.

(3) Manuscript modified to correctly represent the phenomenon.

(1) p3 l27: typo: based -> based on

(2) As pointed out, this was a typo.

(3) It has been corrected in the revised manuscript.

(1) p4 l25: "SANOMA will express NO in number density to accommodate chemical models" Why is number density over altitude better suited to accommodate chemical models than vmr over pressure level?

(2) The idea here was that certain models (e.g. WACCM) requires its constituents in number density. However, a more proximate reason was that the old model, NOEM, was in number density and hence comparison would be greatly facilitated if SANOMA was also in the same units.

(3) This point has been clarified.

(1) p8 l 6: the PC's mentioned here are not included in Eq. 1. It would be clearer to note that the PCs were replaced by polynomial fits to the geophysical parameters Kp, delta, log(F10.7) in Marsh et al., 2004, represented in Eq 1 by f1,f2, and f3.

(2) This connection was missing from the original manuscript.

(3) Added connection that Marsh et al. used polynomial fits to the geophysical parameters as substitutes for PC1-PC3.

(1) p10 l9: Have auto-correlations been considered in the regression? This would be important since explained variances might be overestimated otherwise (see, e.g. Hendrickx et al., GRL, 44, 2017)

(2) The reviewer raises a very good point here. Autocorrelation in the residual was tested using the Ljung-Box Q test. It was found to be present, although it is very low. The correlogram in Fig. 1 attached to this response is an example showing the autocorrelation in the residuals at 70°S and at an altitude of 102km, as a function of the time lag. The correlogram in Fig. 2 attached to this response corresponds to a random data set. We can see that auto-correlation in the residuals is only slightly higher than autocorrelation characteristic of a random time series. Moreover, P-values lower than 0.01 at high latitudes above 95km, as well as below 95km at all latitudes, indicate that there is no strong evidence to reject the null hypothesis that the residuals are not autocorrelated in these regions. (This can slightly vary according to the lag into consideration.) For these reasons, autocorrelation in the residuals can be considered as reasonable. It would of course be good to minimise it anyway, but we could not apply the Cochrane-Orcutt procedure to correct our NO data set, as it was done by Hendrickx et al. (2017). As we understand it, this procedure does not deal with long-term correlations. Koen et al. (2017) could use it in their study because they considered deseasonilised anomalies, hence short term variations only. The SMR data set is characterised by an uneven temporal sampling, with only four measurement days a month on an irregular basis. This makes the use of this procedure inappropriate in the

context of our study. Moreover, considering autocorrelation is less crucial than in the study by Koen et al. (2017), where the conclusions were built upon R-squared values. In our case, the coefficients of determination are not the actual result of the study, but rather a tool that indicates how well the model represent the data. Nevertheless, we agree that it is important to estimate these coefficients as accurately as possible. Finding the best way to do that requires further investigation and we keep that in mind for a next version of the model.

(1) p 11 l11: "R-sqared" coefficient of determination

(3) This point has been clarified.

(1) p11 l8: Why aren't the mean differences between SANOMA and SMR zero in a given bin? This would be expected from a simple multilinear regression.

(2) The function of the figure is to indicate that the differences between SANOMA and SMR are extremely small in comparison to the measured values (10E7 vs. 10E-7). We assume this error to be negligible and due to numerical error in creating the model.

(3) A passage clarifying the significance of the figure has been added to the manuscript.

(1) p13 l10: "SANOMA includes no dynamics". Since SANOMA is an empirical model it does not include any physical process.

(2) Yes, as an empirical model, SANOMA does not intend to reproduce any physical processes. What we meant was that it does not include any parameter to account for dynamical processes, as it does with Kp, delta and F10.7cm to account for the geomagnetic activity, the seasonal variations and the solar activity, respectively.

(3) The manuscript has been modified to clarify what is meant by "no dynamics"

(1) p13 l11: "This could also be due to a lower signal-to-noise ratio in the measurements at lower altitudes." Isn't the measurement error already accounted for in Eq 5?

(2) Yes, Equation 5 accounts for measurement error, but since the measured NO number densities are comparatively small, a large relative error is unsurprising.

(3) We have changed the text in order to make that point clearer.

(1) p14 Table 1: Isn't ACE measuring at altitudes < 70 km as well?

(2) The reviewer is correct to point out this mistake in the table. However, we found that including the altitude range bears no significance for this particular study since the assumption is that all cover the upper mesosphere and lower thermosphere. The sole exception to this is the ACE-FTS instrument, whose maximum altitude is around 108 km but that is mentioned in the appropriate section.

(3) Removed altitude range column.

(1) p14 Table 1: Same for MIPAS. Also, the MIPAS latitude range should be 90S - 90N.

(3) MIPAS' latitude range has been corrected.

(1) p 15 Fig 12: I'm puzzled about the NOEM results, showing minimum polar summer NO concentrations in 2012-2014, i.e., well after the solar cycle minimum (2009-2010). How is this possible if NOEM uses F10.7 and Kp (both having lower values in solar min)?

(2) This is a very interesting and correct observation: the NOEM-simulated NO does indeed show a minimum between 2012-2014 instead of the expected 2009-2010. The reason for this seems to lie in the EOFs of NOEM. Figure 2 d) in the paper by Marsh et al. (2004) reveals that EOF 3 is positive between roughly -60 and +60 magnetic latitude, but negative close to the poles. The plotted time series in Figure 12 of our paper is taken at -70 degrees magnetic latitude and 102 km in altitude. At this location, the 3rd EOF has a value of roughly -2.5. Since this value is negative, a decrease in F10.7cm flux will result in an increase in modeled NO and vice versa. Concerning the negative values close to the poles, Marsh et al. (2004) wrote: "The third EOF shows an unexpected decrease at latitudes greater than 50° that also appears related to solar UV

variability. A possible explanation is that this may again be related to an increase in the suppression of discrete auroral arcs as solar fluxes increase toward solar maximum. Alternatively, solar cycle induced variations in chemical composition or temperature of the lower thermosphere may be indirectly affecting NO production and loss and so the distribution of NO. " In short, the negative values were unexpected. We suspect that this negative value in EOF3 can lead to the minimum NO modeled by NOEM to occur at another time than 2009. For comparison, we can have a look at NOEM-modeled NO at the same dates as in Figure 12, but for the equatorial latitude band and the same height, see the attached Fig. 3. It shows that at the equator, where EOF 3 is positive, the time of lowest solar activity does coincide with the lowest NOEM NO. Therefore we believe that the NOEM minimum in Figure 12 is indeed caused by the negative value of EOF 3 in the study by Marsh et al. (2004) at that particular latitude.

(1) p18 l5 : The given reference is correct for MIPAS but for the NO data version mentioned here Bermejo-Pantaleon et al, 2011 would be more adequate. It is unfortunate that this study uses data version v5r 620 since it only covers the years 2010-2012. The newer version v5r 622 (available, for instance, at http://mesospheo.fmi.fi/data_service.html) covers the entire period 2005-2012.

(2) The new version of the data has been downloaded and used for all the analyses in the updated manuscript.

(3) The input data for the MIPAS section has been changed to v5r 622, meaning that the ensuing figures, values, and following conclusions have all been modified.

(1) p19 l2: "with a minimum of -60% at 115 km around 70 deg". It looks more like a minimum of -30 to -40% at 115 km in Fig. 22.

(2) This comment is no longer valid since we have revised this section due to the use of the newer v5r 622 MIPAS NO data.

(1) p21 Table 2: caption: "the mean of the mean percent difference". Do you mean

the mean percent difference averaged over the respective latitude bands? Further: "between the various instruments and a) NOEM, b) SMRNOEM, and c) SANOMA". Shouldn't it be the other way round, i.e., model - instrument?

(2) We calculate the percent difference for each latitude band, and then calculate the average of these three. The difference is indeed model-instrument, not instrument-model as the caption might suggest.

(3) Changed caption to clarify the meaning of the values in the table.

(1) p21 l4: Since SMR measures both day and night time NO, a positive difference compared to the daytime measuring instrument SCIAMACHY was expected, but SANOMA shows a positive difference in comparison to SOFIE and ACE as well". SOFIE and ACE are solar occultation instruments. A diurnal sampling bias can therefore no be excluded when comparing to instruments that measure at both day and nighttime (like SMR or MIPAS).

(2) This is true.

(3) This point has been clarified in the revised manuscript.

(1) p21 l12-15: I wouldn't say that MIPAS measurements should be treated with care, particularly because MIPAS shows small absolute difference with respect to SANOMA (and hence SMR), as well as high R-squared values (Table 1). It is likely that the apparently different behavior of MIPAS compared to other datasets is related to the shorth time coverage of the MIPAS data version used here (2010-2012). It would be very interesting to check this by using the newer version v5r 622 (2005-2012).

(2) This comment is no longer valid since the v5r 622 MIPAS NO has been used and the ensuing conclusions have changed.

(1) p22 l5-7: see above.

(2) See above.

Reviewer 2 comments:

(1) How was is the regression model built? We have the equation:

NO(lamda, h, t) = Kp(t)*c1 + dec(t)*c2 + log(F10.7(t))*c3 + com1(Kp(t),dec(t))*c4 + com2(Kp(t),dec(t))*c5 + C

Firstly it is not clear how the authors formed com1 and com2. The paper says that they were a result of iteration, but there is no physical explanation for them or clear explanation of the iteration process. They do, however, introduce autocorrelation with all except term #3. I am worried that there are two terms which both have a linear (lagged) Kp term as well as linear/sin terms depending on solar declination in addition to the linear terms in #1 and #2 - without any explanation.

(2) Please refer to the response to the next comment.

(1) The final number of terms in #4 and #5 also depend on solar declination in a way that is not explained clearly. Why for summer solstice conditions are there a maximum number of lagged days (11)? Would it not be natural to assume a lag is needed for winter conditions when the lifetime of NO is larger and thus there might be build-up? Why was the Kp lag not built into the first term of the regression (and same for solar declination)? What is the physical meaning of having these extra Kp & dec dependent terms?

(2) This was a very constructive comment, which warranted a closer inspection of the compound indices. We found an error in how they were built which has been corrected. The basic idea now is that com1 and com2 vary linearly with solar declination, see the updated equations in the manuscript. Com1 is zero during northern hemispheric summer and reaches a maximum during Arctic winter. Meanwhile com2 is zero during southern hemispheric summer, reaching its maximum during Antarctic winter. This means that com1 accounts for NO produced by energeric particle precipitation during previous days during northern winter, the time when the polar air mass receives least

sunlight and dissociation of NO by sunlight is at a minimum. In essence, the closer to winter solstice we get, the longer back in time com1 goes in integrating over NO from previous days. We know that in the absence of sunlight, NO can persist for longer periods of time and therefore we can have a build-up of NO in the dark polar air mass. The same rationale applies for com2, but for the southern hemisphere. The newly added figure in the manuscript show the coefficients of the multivariate linear fits corresponding to com1 and com2. It confirms that com1 and com2 influence the measured NO only in the northern, and southern hemispheres, respectively. They act as a "memory" of the NO build-up from previous days during polar winter. Figure 11 b) and c) demonstrate that these added indices do in fact sustantially improve SANOMA in the polar regions.

(3) A detailed description of the corrected indices, including comments on their physical meaning, has been added to the revised version of the manuscript.

(1) The actual regression coefficients "c1, c2,. . .c5" are not given (First 3 were plotted in Figure 8, I didn't notice remarks on c4 and c5) and thus the equation can not be used. Please note that many readers will not have access to Matlab statistics toolbox used for the analysis.

(2) As mentioned at a later point in the paper, all the coefficients are available at odin.rss.chalmers.se.

(3) We have added a sentence at the point of introduction of the coefficients to clarify that these are available online. Moreover, a figure representing the coefficients c4 and c5 (Fig.9), was added to the manuscript, together with the corresponding detailed description.

(1) Other comments: Figure 2: This is plot of the F10.7 time series, clearly taken directly. Although the reference to the website is given, please plot the data yourself. This is a very simple figure with the actual monthly means and a running mean. The future prediction (red line) is not necessary for this paper.

(3) We have replaced the figure with a new one in which we plotted the relevant F10.7cm time series data ourselves.

(1) The indices you use here are not "space weather" indices, but rather geomagnetic and solar indices. The term space weather has a very specific meaning relating to impacts on technology (and these indices can be used for those), but as we are looking at impacts on the Earth's atmosphere we should talk about geomagnetic and solar indices.

(2) Yes, we were using the term "space weather" rather inaccurately.

(3) This has been changed to "geomagnetic and solar indices".

(1) The AE index (Auroral Electrojet) is mentioned in the same section, but this is not used anywhere after that, only Kp is mentioned.

(2) If we don't mention the AE index in the paper, some readers could think that we should have considered the AE index instead of the Kp index. The function of mentioning the AE index is to indicate that we have tested it but found that the Kp index explains a larger portion of the variance of the Odin SMR NO measurements.

(3) Added clarification that AE is not used for the final model, but we did consider it.

Please also note the supplement to this comment:
https://www.atmos-chem-phys-discuss.net/acp-2018-39/acp-2018-39-AC1-supplement.pdf

[Figure]

**ACF for residuals (SANOMA-SMR) for 70S, 102 km, all indeces**

**Fig. 1.**

[Figure]

**ACF for a randomly generated time series**

Fig. 2.

[Figure]

**Equatorial NO**

Legend: SANOMA, NOEM

**Fig. 3.**

**Supplement:**

[revised manuscript text omitted]

---

## Author Response (AR2)

We would like to thank once again the two referees, Dr. Bernd Funke and an anonymous referee, as well as the editor, Dr. William Ward, for their efforts to help improve our paper. This response addresses the remaining points. Below, the reviewer's comments are marked with (1), the author's response with (2), and the change in the manuscript with (3).

Reviewer 1 comments: accepted as is

Reviewer 2 comments:

(1) The authors did not respond to my comment about autocorrelation, but as reviewer #1 queried about it too, I was able to read the response there. There is no mention about this issue is the present text and the authors indicate that they will keep this in mind "for next version of the model". However, I still think it would be worth adding a sentence to the present manuscript, based on the work the authors did to respond to reviewer #1. I.e. it's present, but low.

(2) We agree that adding some reflection on the autocorrelation issue will help improve and clarify the article.

(3) We have added a short reflection on the autocorrelation in the new manuscript on page 14.

(1) Page 2, line 25: "NOx" (or even NOy) descents into the stratosphere, not NO as the descending NO gets converted to NO2 in mesosphere. NO2 is photolysed back to NO.

(2) The reviewer correctly points out an inaccuracy in the manuscript.

(3) "NO" changed to "NOx".

(1) Figure 2: Units of sfu are missing, these should be added to the caption for example.

(2) Another place to make the manuscript clearer.

(3) Added units for sfu.

(1) Page 21, line 5: I understand this is speculation, but these altitudes are a bit high for the polar vortex.

(2) We agree that the altitudes in question are a bit high for the polar vortex. However, the data does suggest a difference in the performance of SANOMA between the two hemispheres. We cannot quite put a finger on this observed difference.

(3) Changed "more stable polar vortex" to "more stable dynamics".

(1) Page 22, line 6: "profits". I think you mean "benefits".

**ACPD**
(2)

- (3) Changed "profits" to "benefits".
- (1) Figures: Particularly the multi-panel figures should be larger in final version.

(2)

(3) All figure sizes have been increased.

[revised manuscript text omitted]